# Cooperation of mitochondrial and ER factors in quality control of tail-anchored proteins

Verena Dederer[1], Anton Khmelinskii[1,2], Anna Gesine Huhn[1], Voytek Okreglak[3], Michael Knop[1,4], Marius K Lemberg[1]*

[1]Centre for Molecular Biology of Heidelberg University (ZMBH), DKFZ-ZMBH Alliance, Heidelberg, Germany; [2]Institute of Molecular Biology (IMB), Mainz, Germany; [3]Calico Life Sciences LLC, South San Francisco, United States; [4]Cell Morphogenesis and Signal Transduction, German Cancer Research Center (DKFZ), Heidelberg, Germany

**Abstract** Tail-anchored (TA) proteins insert post-translationally into the endoplasmic reticulum (ER), the outer mitochondrial membrane (OMM) and peroxisomes. Whereas the GET pathway controls ER-targeting, no dedicated factors are known for OMM insertion, posing the question of how accuracy is achieved. The mitochondrial AAA-ATPase Msp1 removes mislocalized TA proteins from the OMM, but it is unclear, how Msp1 clients are targeted for degradation. Here we screened for factors involved in degradation of TA proteins mislocalized to mitochondria. We show that the ER-associated degradation (ERAD) E3 ubiquitin ligase Doa10 controls cytoplasmic level of Msp1 clients. Furthermore, we identified the uncharacterized OMM protein Fmp32 and the ectopically expressed subunit of the ER-mitochondria encounter structure (ERMES) complex Gem1 as native clients for Msp1 and Doa10. We propose that productive localization of TA proteins to the OMM is ensured by complex assembly, while orphan subunits are extracted by Msp1 and eventually degraded by Doa10.

DOI: https://doi.org/10.7554/eLife.45506.001

*For correspondence:
m.lemberg@zmbh.uni-heidelberg.de

## Introduction

Correct localization of proteins is essential to ensure their functionality and to establish the identity of individual cellular organelles. Most proteins are synthesized in the cytosol and targeted to subcellular compartments either co-translationally or after their synthesis is completed, post-translationally (*Aviram and Schuldiner, 2017*; *Hegde and Keenan, 2011*; *Wasilewski et al., 2017*). The signal recognition particle (SRP) recognizes N-terminal signal sequences of ER-targeted nascent polypeptides and recruits the ribosome-nascent chain complex to the Sec61 translocon (*Voorhees and Hegde, 2016*). Membrane proteins that are not recognized by SRP are captured by other cytosolic chaperons, which keep these proteins in an unfolded state guiding them to the ER, mitochondria or peroxisomes post-translationally (*Aviram and Schuldiner, 2017*; *Hegde and Keenan, 2011*; *Wasilewski et al., 2017*). Tail-anchored (TA) proteins are a specific class of membrane proteins that have a single transmembrane (TM) domain at their very C-terminus. They are involved in various cellular processes such as membrane fusion, protein translocation and regulation of apoptosis (*Antonsson, 2001*; *Beilharz et al., 2003*; *Burri and Lithgow, 2004*; *Chen and Scheller, 2001*; *Kalbfleisch et al., 2007*). The GET pathway (guided entry of TA proteins) ensures targeting of TA proteins to the ER (*Hegde and Keenan, 2011*). In brief, a cytosolic pre-targeting complex comprising Sgt2, Get4, and Get5 captures the TM segment of a TA protein after it emerges from the ribosome and loads it onto Get3 (*Mariappan et al., 2010*; *Mateja et al., 2009*; *Schuldiner et al., 2008*).

This cytosolic ATPase targeting factor is subsequently recruited to the ER membrane via the Get1-Get2 receptor complex where, after ATP hydrolysis, the TA protein is inserted into the membrane (*Mariappan et al., 2011*; *Schuldiner et al., 2008*; *Wang et al., 2011*). Recently, the EMC complex (ER membrane protein complex) was shown to ensure ER targeting of a subset of TA proteins in mammalian cells (*Guna et al., 2018*). However, no dedicated pathway targeting TA proteins to the outer mitochondrial membrane (OMM) has been identified so far and it is unclear how they insert into the lipid bilayer (*Vitali et al., 2018*). From in vitro studies, it has been proposed that because of the low ergosterol content of the OMM TA proteins can insert unassisted (*Kemper et al., 2008*).

Given these multiple, possibly overlapping, targeting mechanisms, insertion of TA proteins appears to be intrinsically prone to failure. Moreover, uncontrolled mitochondrial targeting requires protein quality control systems to remove mistargeted and surplus TA proteins. To ensure protein homeostasis (proteostasis), organelle-specific safeguards recognize orphan complex subunits, damaged and mistargeted protein species and direct them for degradation by the ubiquitin-proteasome system (*Juszkiewicz and Hegde, 2018*). In the ER, this is linked to polytopic E3 ubiquitin ligases that, in concert with a diverse set of additional factors, constitute the ER-associated degradation (ERAD) pathway (*Avci and Lemberg, 2015*; *Mehrtash and Hochstrasser, 2018*; *Ruggiano et al., 2014*). In yeast, three major branches of ERAD are centered around the polytopic E3 ubiquitin ligases Hrd1, Doa10 and Asi1/3 (*Bays et al., 2001*; *Carvalho et al., 2006*; *Deak and Wolf, 2001*; *Foresti et al., 2014*; *Khmelinskii et al., 2014*; *Swanson et al., 2001*). While Hrd1 is specific for proteins with misfolded ER luminal or aberrant TM domains, Doa10 primarily mediates degradation of ER proteins with cytosolic lesions (*Carvalho et al., 2006*). The Asi1/3 complex removes misfolded or mislocalized membrane proteins from the inner nuclear membrane, a sub-area of the ER (*Foresti et al., 2014*; *Khmelinskii et al., 2014*). In addition, Doa10 encompasses broader functionality by targeting soluble proteins for proteasomal degradation and was suggested to perform a quality control of proteins that fail membrane insertion (*Ast et al., 2014*; *Kats et al., 2018*; *Maurer et al., 2016*; *Swanson et al., 2001*). In this context, Doa10 triggers degradation of proteins with hydrophobic sequences at either end of the protein, such as hydrophobic C-terminal prodomains of glycosylphosphatidylinositol (GPI)-anchored proteins (*Ast et al., 2014*) or hydrophobic N-termini (*Kats et al., 2018*). For retro-translocation and extraction of membrane-embedded ERAD substrates the required energy is provided by the AAA-ATPase Cdc48 (*Ye et al., 2001*). Cdc48 substrate recognition is mediated through its Ufd1-Npl4 cofactor, which binds ubiquitinated proteins (*Meyer et al., 2000*). While this process is best understood in the ER, analogous extraction pathways have been described for mitochondria, the Golgi complex and peroxisomes (*Avci and Lemberg, 2015*; *Chen et al., 2014*; *Okreglak and Walter, 2014*; *Schmidt et al., 2019*; *Stewart et al., 2012*). Likewise, Ubx2 which was thought to be an ERAD-specific Cdc48 adaptor, has been shown to also bind to the TOM (translocase of the outer membrane) complex, defining a mitochondrial protein translocation-associated degradation pathway (*Mårtensson et al., 2019*). In a related process referred to as mitochondria-associated degradation, Doa1 or Vms1 recruit the Cdc48 complex to OMM-anchored ubiquitinated proteins to target them for proteasomal degradation (*Anton et al., 2011*; *Heo et al., 2010*; *Karbowski and Youle, 2011*; *Wu et al., 2016*). Whereas this pathway is linked to the ubiquitin-proteasome system through the cytoplasmic E3 ubiquitin ligases Rsp5 and Mdm30, an alternative extraction mechanism relies on the AAA-ATPase Msp1 (known as ATAD1/Thorase in humans). Msp1 shows dual localization to the OMM and peroxisomes (*Chen et al., 2014*; *Okreglak and Walter, 2014*). In peroxisomes it has been suggested, that the TA protein Pex15 evades Msp1-dependent extraction by interacting with other peroxisomal proteins such as Pex3 (*Weir et al., 2017*). Deletion of Msp1 leads to stabilization of mistargeted peroxisomal Pex15 or the Golgi v-SNARE Gos1 in the OMM (*Chen et al., 2014*; *Okreglak and Walter, 2014*). In vitro studies suggest that Msp1 is sufficient to extract TA proteins from proteoliposomes (*Wohlever et al., 2017*). Although these reconstitution experiments showed that the Msp1 N-terminal TM domain is dispensable for the dislocation reaction, recently Li *et al.* suggested that a conserved negatively charged aspartate residue in the N-terminal portion of Msp1 facing the inter membrane space (IMS) is important for efficient recognition of Msp1 clients in vivo (*Li et al., 2019*). Moreover, a hydrophobic surface of the AAA-ATPase domain has been implicated in recognition of Msp1 clients (*Li et al., 2019*). However, no clear sequence consensus between all known Msp1 clients exists (*Chen et al., 2014*; *Li et al., 2019*; *Okreglak and Walter, 2014*) and the mechanism how Msp1 distinguishes between mislocalized and properly located TA proteins is ill defined. This, taken together with the

finding that Cis1 can recruit Msp1 to TOM for degradation of stalled import intermediates (*Weidberg and Amon, 2018*), indicates that multiple routes can target proteins for Msp1-mediated dislocation. Furthermore, the fate of TA proteins extracted in the cytoplasm by Msp1 is unclear. To address these issues, we screened for additional factors important for clearing mistargeted TA proteins from the OMM. Our findings outline a previously unanticipated role of the ER-resident E3 ubiquitin ligase Doa10 as a major regulator in the control of OMM TA protein targeting.

## Results

### Screen for factors involved in turnover of mistargeted Pex15Δ30

To identify factors involved in degradation of mistargeted TA proteins in yeast, we studied a Pex15 variant lacking the 30 C-terminal residues (Pex15Δ30). Whereas wild type (wt) Pex15 localizes to peroxisomes, Pex15Δ30 constitutively mislocalizes to the OMM and exhibits Msp1-dependent turnover (*Okreglak and Walter, 2014*). Pex15Δ30 N-terminally tagged with the circular permuted cp8 superfolder GFP variant (referred to GFP hereafter), which shows less cytoplasmic background due to improved proteasomal degradation (*Khmelinskii et al., 2016*), localized predominantly to mitochondria and only to a minor extent to peroxisomes in wt and *msp1Δ* cells (*Figure 1—figure supplement 1A and B*). To be able to measure Pex15Δ30 turnover in high-throughput in different mutants, we fused Pex15Δ30 to a tandem fluorescent protein timer (tFT) consisting of the fast-maturing superfolder GFP (sfGFP) and the slower-maturing mCherry (*Figure 1A*) (*Khmelinskii et al., 2012*). A population of newly synthesized fusion proteins exhibits first sfGFP fluorescence and acquires mCherry fluorescence over time. This property allows use of the mCherry/sfGFP ratio as a measure of protein turnover, whereby the ratio increases as a function of protein half-life (*Khmelinskii et al., 2012*). We introduced the tFT-Pex15Δ30 construct into an arrayed collection of non-essential yeast gene deletion strains (*Winzeler et al., 1999*) using high-throughput strain construction (*Tong and Boone, 2006*) and measured the mCherry and sfGFP fluorescence of colonies grown on agar (*Figure 1A*) (*Khmelinskii et al., 2014*; *Khmelinskii et al., 2012*). Because the tFT readout can be affected by perturbations unrelated to protein turnover, for example changes of mitochondrial physiology (*Khmelinskii et al., 2012*), we performed a control screen using tFT-tagged Tom5 (*Figure 1A*). Tom5 is a stable TA protein subunit of the preprotein translocase TOM. Gene deletions that significantly increased tFT-Pex15Δ30 abundance and stability but did not affect tFT-Tom5 were considered as hits (*Figure 1B*, *Supplementary file 1*). According to this analysis, tFT-Pex15Δ30 was stabilized in *msp1Δ* as has been observed before using GFP-tagged constructs (*Chen et al., 2014*; *Okreglak and Walter, 2014*). Moreover, deletion of *GET3*, which was previously linked to trafficking of full-length Pex15 and shows a negative genetic interaction with *msp1Δ* (*Costanzo et al., 2010*; *Okreglak and Walter, 2014*), caused a significant increase in sfGFP level and mCherry/sfGFP ratio. tFT-Pex15Δ30 was also significantly stabilized in the absence of the ER-resident P-type ATPase Spf1, the pheromone regulated membrane protein Prm1 and, interestingly, the ER-resident E3 ubiquitin ligase Doa10, its associated E2 ubiquitin-conjugating enzyme Ubc7 and its tethering factor Cue1 (*Figure 1B*). A mutant lacking Ubc6, the second E2 ubiquitin-conjugating enzyme mediating protein degradation together with Doa10 (*Weber et al., 2016*), was not present in the yeast strain collection used for screening. To validate these results, we manually deleted each hit in a haploid strain expressing tFT-Pex15Δ30 and measured strain fluorescence with flow cytometry. Strains *lacking MSP1, DOA10, UBC7, CUE1, GET3* or *SPF1* consistently showed increased sfGFP fluorescence and mCherry/sfGFP ratios compared to wt (*Figure 1C*). Deletion of *UBC6* also stabilized tFT-Pex15Δ30. In contrast, we did not observe stabilization of tFT-Pex15Δ30 in *prm1Δ* cells and excluded this mutant from further analysis. Next, we used fluorescence microscopy to examine the localization of tFT-Pex15Δ30 in each mutant. In *msp1Δ, doa10Δ, ubc6Δ, ubc7Δ,* and *get3Δ* mutant strains tFT-Pex15Δ30 accumulated in mitochondrial structures (*Figure 1—figure supplement 1B and C*).

### *Disparate effects of spf1Δ and get3Δ on Pex15Δ30 turnover*

Whereas ablation of Get3 or the Doa10 complex lead to accumulation of Pex15Δ30 in mitochondria, deletion of *SPF1* led to pronounced ER localization (*Figure 2A* and *Figure 1—figure supplement 1C*). This finding is consistent with a previous report demonstrating that mitochondrial TA proteins mislocalize to the ER in *spf1Δ* strains (*Krumpe et al., 2012*). This phenotype is thought to be caused

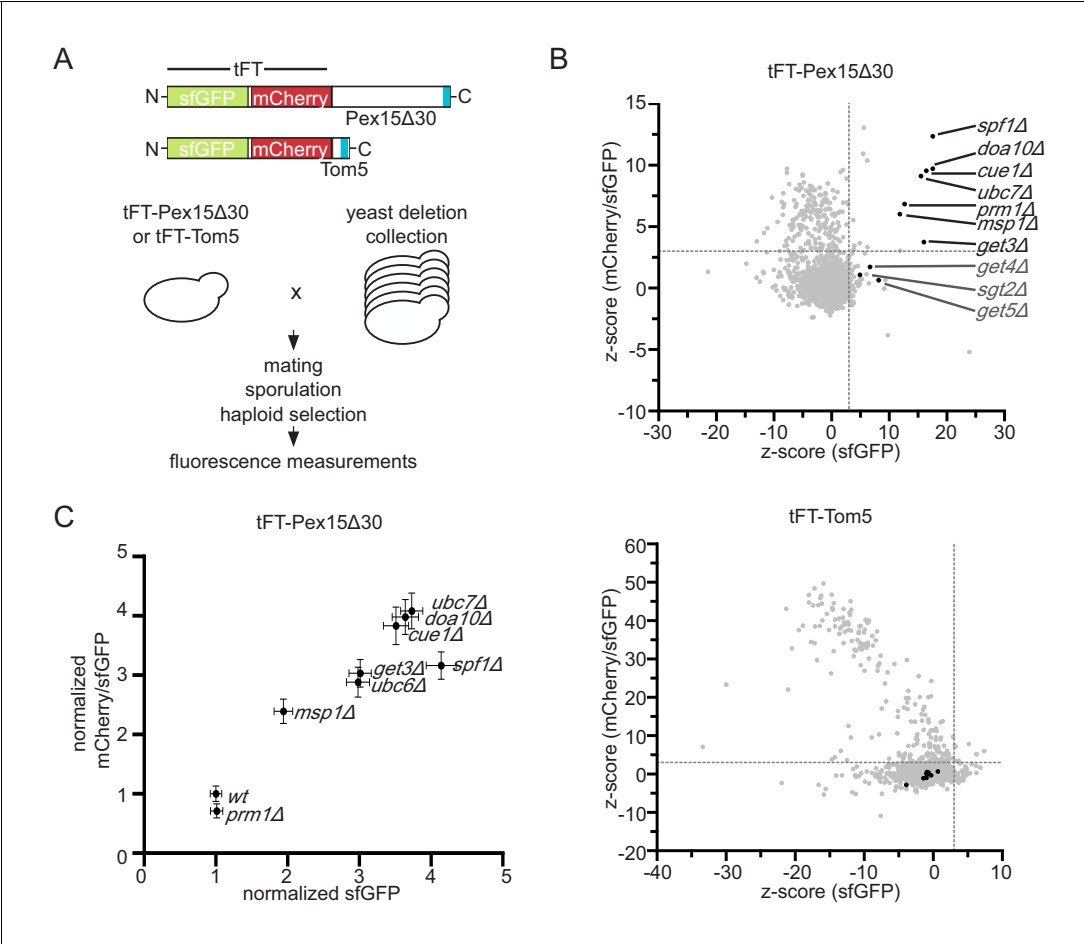

**Figure 1.** Genome-wide screen to identify factors stabilizing Pex15Δ30 TA protein. (**A**) Overview of the tFT-Pex15Δ30 and tFT-Tom5 TA protein reporters. Tail-anchor is highlighted in blue. Strains, expressing either tFT-Pex15Δ30 or tFT-Tom5 from the *TEF1* promoter were crossed into the yeast non-essential gene deletion collection (*Winzeler et al., 1999*) using automated mating and selection procedure. sfGFP and mCherry fluorescence was acquired from arrayed colonies grown on agar (n = 4). (**B**) sfGFP signal and mCherry/sfGFP ratio of the tFT-Pex15Δ30 and tFT-Tom5 reporters of each mutant shown as z-score (which resembles the standard deviations from the mean a data point is). Mutants with z-scores > 3 for sfGFP and mCherry/sfGFP ratio (at 5% false discovery rate) for Pex15Δ30 and not affecting Tom5 are highlighted in black. Dashed gray lines indicate the thresholds. *GET* mutants below the threshold are highlighted in dark gray. (**C**) Flow cytometry validation of generated yeast mutants as indicated. Mean sfGFP intensities and mCherry/sfGFP ratios normalized to wt (n = 4, ± SEM).

DOI: https://doi.org/10.7554/eLife.45506.002

The following figure supplement is available for figure 1:

**Figure supplement 1.** Colocalization of Pex15Δ30 reporter with cellular markers.

DOI: https://doi.org/10.7554/eLife.45506.003

by altered ER ergosterol levels in *spf1Δ*, mimicking conditions of the OMM. Interestingly, in addition to the ER localization, the overall levels of the tFT-Pex15Δ30 reporter increased in the *spf1Δ* mutant (*Figure 1C*). This indicates that *spf1Δ* cells cannot efficiently degrade Pex15Δ30. Western blot analysis revealed a characteristic 26 kDa remnant of sfGFP that persists vacuolar degradation in *spf1Δ* cells (*Figure 2B*) (*Khmelinskii et al., 2016*; *Shintani and Klionsky, 2004*). Monitoring tFT-Pex15Δ30 localization in *spf1Δ* strains by microscopy revealed that this accumulation is likely caused by transport of ER-inserted protein through the secretory pathway. Whereas in the ER only the fast maturing sfGFP and no mCherry signal can be detected, the latter is observed in the vacuole (*Figure 1—figure supplement 1C*). However, if Pex15Δ30 is, in addition, directly targeted into the vacuole or multivesicular body membrane is not resolved yet.

To understand if deletion of *GET3* stabilizes Pex15Δ30 directly or indirectly, we investigated the effect of other components of the GET pathway on cellular levels of tFT-Pex15Δ30. Of note, in the

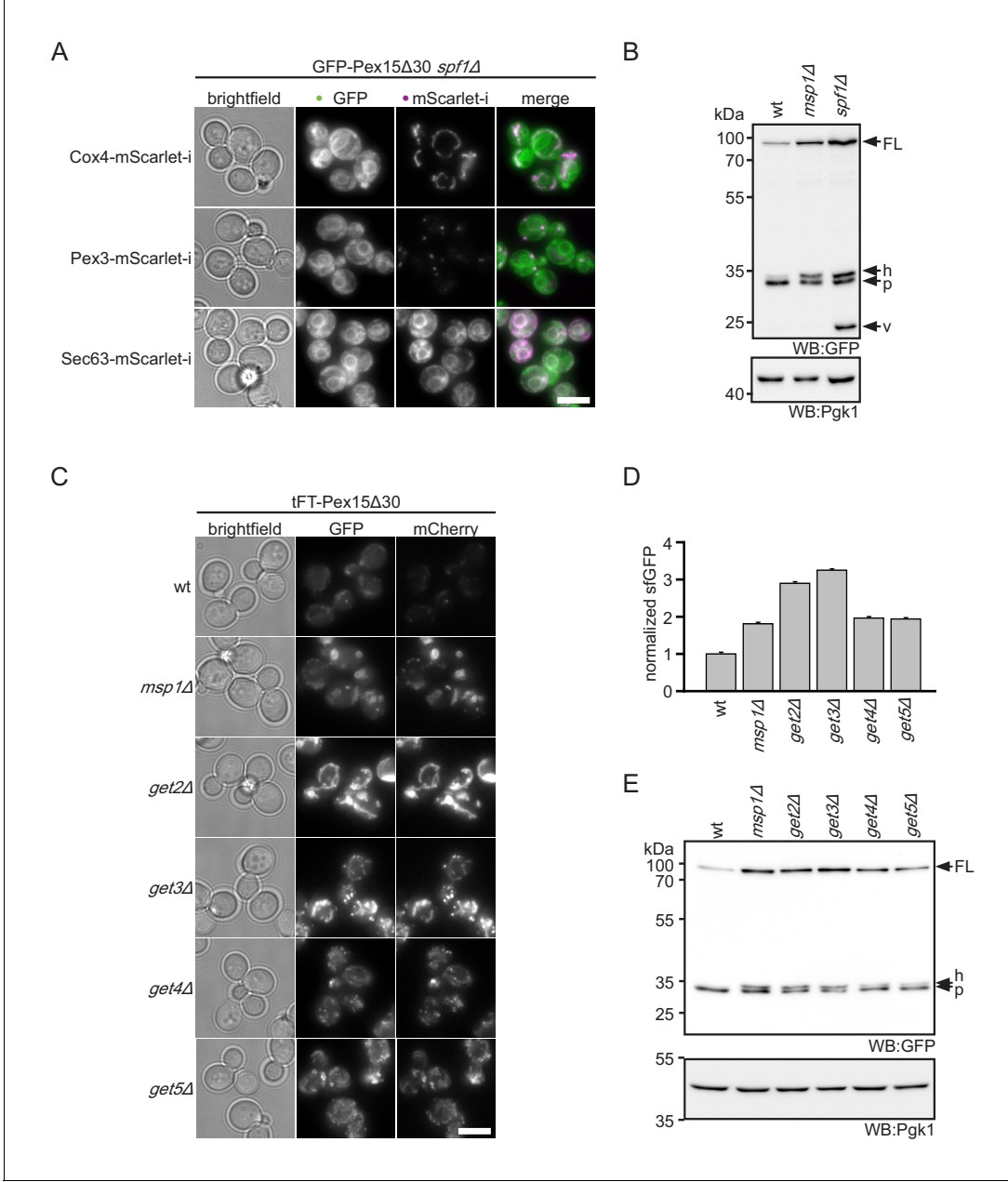

**Figure 2.** ER insertion and mitochondrial accumulation impedes efficient Pex15Δ30 degradation in *spf1Δ* and *get3Δ*. (A) Microscopy analysis of *spf1Δ* strains expressing GFP-Pex15Δ30 from the *TEF1* promoter. Co-expression of chromosomally tagged cellular marker proteins: Cox4-mScarlet-i, mitochondria; Pex3-mScarlet-i, peroxisomes; Sec63-mScarlet-i, ER. Images are adjusted for optimal display range. Colocalization is colored in white in merge (green - GFP, magenta - mScarlet-i). Scale bar: 5 μm. (B) Western blot (WB) analysis of log phase grown wt, m*sp1Δ*, and *spf1Δ* strains expressing tFT-Pex15Δ30. Probing with anti-GFP antibody detects full length protein (FL) and degradation resistant tFT intermediates (*Khmelinskii et al., 2016*): h, SDS-induced mCherry hydrolysis product; p, proteasomal-degradation resistant fragment; v, vacuolar degradation-resistant fragment. Pgk1 serves as loading control. (C) Microscopy analysis of *get2Δ, get3Δ, get4Δ, and get5Δ* strains expressing tFT-Pex15Δ30 compared to wt and *msp1Δ*. Scale bar: 5 μm. (D) Flow cytometry GFP measurement of strains from (C) normalized to wt (n = 3, ± SEM). (E) WB analysis of log phase grown strains from (C) with detection of degradation resistant tFT intermediates.

DOI: https://doi.org/10.7554/eLife.45506.004

The following figure supplement is available for figure 2:

**Figure supplement 1.** Steady state level of tFT-Pex15Δ30 in yeast mutants.

DOI: https://doi.org/10.7554/eLife.45506.005

screen, mutants lacking the GET components Sgt2, Get4 and Get5 showed effects just below the threshold (*Figure 1B*). Similar to *get3Δ*, deletion of the ER membrane receptor subunit Get2 or the cytosolic pre-targeting factors Get4 and Get5 resulted in increased levels of the tFT-Pex15Δ30 and accumulation in mitochondria (*Figure 2C and D*). Western blot analysis showed an increase of tFT-Pex15Δ30 level in a similar range as *msp1Δ* for all *GET* mutants (*Figure 2E*). Of note, the characteristic 26 kDa vacuolar tFT degradation intermediate observed for *spf1Δ* was not seen in *get3Δ* cells (*Figure 2—figure supplement 1*). Taken together, these results indicate that the *GET* mutants interfere with the tFT-Pex15Δ30 reporter by blocking canonical ER targeting function (involving the ER-resident membrane receptor Get2). Hence, Pex15Δ30 accumulation in mitochondria is most likely a secondary consequence of altered TA protein homeostasis in GET mutants (see below).

## The Doa10 E3 ligase is a major factor in Pex15Δ30 turnover

Our screen revealed that Doa10 is important for degradation of Pex15Δ30 (*Figure 1B*). Rescreening of the tFT-Pex15Δ30 reporter against a library of mutants in the ubiquitination machinery confirmed that Ubc6, Ubc7 and Doa10 are the only significant hits (*Figure 3—figure supplement 1A and B*). Fluorescence measurements with flow cytometry and cycloheximide chase experiments independently revealed reduced turnover of the tFT-Pex15Δ30 reporter in the *DOA10* mutant (*Figure 1C* and *Figure 3—figure supplement 1C*). Notably, tFT-Pex15Δ30 levels were higher in *doa10Δ* compared to *msp1Δ*, and further increased in the *msp1Δdoa10Δ* double mutant (*Figure 3A and B*). Different to an *msp1Δget3Δ* double deletion, which shows a fitness defect (*Chen et al., 2014*; *Okreglak and Walter, 2014*), fitness of the *msp1Δdoa10Δ* mutant was indistinguishable from wt (*Figure 3—figure supplement 1D*), excluding a general toxicity effect and pointing towards a direct role of Doa10 in Pex15Δ30 degradation. Consistent with ubiquitin-dependent degradation, addition of proteasome inhibitors delayed turnover of the Pex15Δ30 reporter (*Figure 3—figure supplement 1E*). Moreover, double deletion of *MSP1* and *DOA10* showed an additive stabilization effect of Pex15Δ30 (*Figure 3A*), suggesting that the two proteins do not act in a linear pathway.

Doa10 is a major player of ERAD-C (*Mehrtash and Hochstrasser, 2018*; *Ruggiano et al., 2014*). However, we observed only mitochondrial accumulation of tFT-Pex15Δ30 in *doa10Δ* (*Figure 3C*), indicating that it acts on pre-inserted TA protein reporter. Moreover, Pex15Δ30 turnover was not affected in the absence of Hrd1 and Asi1/3, the other two ERAD E3 ligases (*Figure 3—figure supplement 1A,B and C*). An alternative degradation route for Pex15Δ30 might be vacuolar degradation. In line with this, western blot analysis of tFT-Pex15Δ30 in *doa10Δ* revealed accumulation of the vacuolar tFT degradation intermediate at 26 kDa (*Figure 3C* and *Figure 2—figure supplement 1*) (*Khmelinskii et al., 2016*; *Shintani and Klionsky, 2004*). Interestingly, additional deletion of *MSP1* decreased abundance of this fragment suggesting that mitochondria act as a 'sink' for mistargeted tFT-Pex15Δ30. Along this hypothesis, the vacuolar sfGFP fragment was absent in the *get3Δ* mutant, supporting the idea that Get3 is required for membrane insertion and subsequent transport of tFT-Pex15Δ30 to the vacuole (*Figure 2—figure supplement 1*). Since in *doa10Δ* no prominent ER localization is observed (*Figure 3—figure supplement 1B*), we assume that tFT-Pex15Δ30 only very transiently exists in the ER. Next, we asked whether under conditions where ER targeting of TA proteins is distorted, also Doa10 activity is compromised. To this end, we studied turnover of the Doa10-dependent ERAD substrate Ste6* (*Stolz et al., 2010*). Interestingly, upon *GET3* deletion Ste6* showed a minor stabilization (*Figure 3—figure supplement 1F*). This may be either caused by mistargeting of the E2 ubiquitin conjugating enzyme Ubc6 leading to a reduction of Doa10-dependent ubiquitination (*Weber et al., 2016*) or cytoplasmic accumulation of TA proteins saturating Doa10 activity. Over all, our results show that the E3 ubiquitin ligase Doa10 plays a prominent role in turnover of the mistargeted TA reporter Pex15Δ30 whereas the GET pathway plays an indirect role via maintaining the general TA protein homeostasis including but not limited to Ubc6.

## Doa10 targets cytoplasmic Pex15Δ30 for degradation to prevent mitochondrial mistargeting

Doa10 and Msp1 localize to two different cellular compartments, the ER and the OMM, respectively. This poses the question in which order these factors interact with the tFT-Pex15Δ30 reporter. We hypothesized that Doa10 either acts directly on newly synthesized tFT-Pex15Δ30, thus preventing its insertion into the OMM, or after extraction from it, whereas Msp1 deals only with the fraction that is

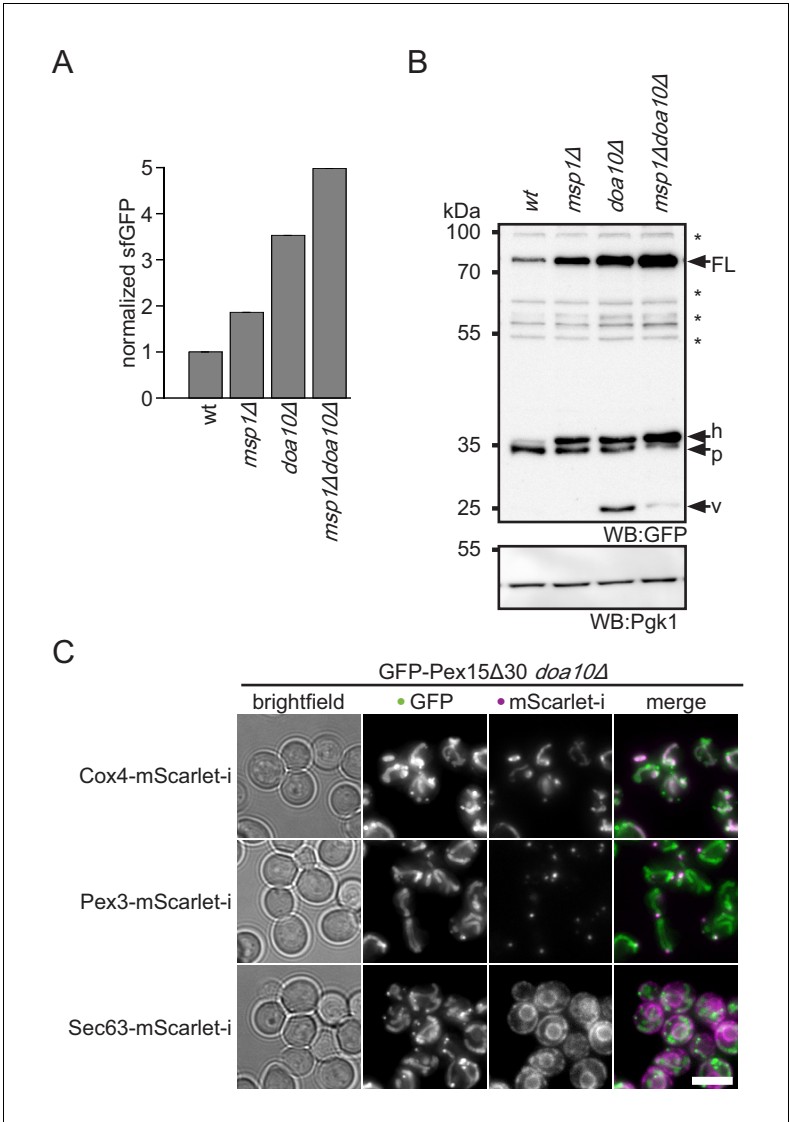

**Figure 3.** Doa10 triggers degradation of cytoplasmic Pex15Δ30. (**A**) Flow cytometry measurements of wt, *msp1Δ*, *doa10Δ* and *msp1Δdoa10Δ* strains expressing tFT-Pex15Δ30. Mean GFP intensities normalized to wt (n = 4, ± SEM). (**B**) Western blot (WB) analysis from log phase grown yeast of (**A**) with full-length tFT-Pex15Δ30 (FL) and degradation resistant tFT intermediates: h, SDS-induced mCherry hydrolysis product; p, proteasomal-degradation resistant fragment; v, vacuolar degradation-resistant fragment. Unspecific bands are marked with star. Pgk1 serves as loading control. (**C**) Microscopy analysis of *DOA10* deletion strains co-expressing GFP-Pex15Δ30 and mitochondrial marker protein Cox4-mScarlet-i, peroxisomal marker protein Pex3-mScarlet-i or ER marker protein Sec63-mScarlet-i. Colocalization is highlighted in white in merge image. Images are shown with optimal display range. Scale bar: 5 μm.

DOI: https://doi.org/10.7554/eLife.45506.006

The following figure supplement is available for figure 3:

**Figure supplement 1.** Targeted screen in ubiquitin-proteasome system mutants emphasizes role of Doa10 in Pex15Δ30 turnover.

DOI: https://doi.org/10.7554/eLife.45506.007

mistargeted to mitochondria. According to this model, cytoplasmic accumulation of Pex15Δ30 in *doa10Δ* strains enhances mitochondrial mistargeting and therefore saturates Msp1-dependent extraction. In order to test this, we overexpressed Msp1 in *doa10Δ* from a galactose inducible promoter (*Figure 4—figure supplement 1A*). To investigate the localization of overexpressed Msp1 we

tagged it at the C-terminus with mNeonGreen. Under galactose induction for 4 hr Msp1 predominantly localizes to mitochondria (*Figure 4—figure supplement 1B*). Galactose induction of Msp1 in the *doa10Δ* background did not affect cellular tFT-Pex15Δ30 level, consistent with the requirement of Doa10 for degradation (*Figure 4A*, *Figure 3—figure supplement 1C*, and *Figure 4—figure supplement 1C*). Fluorescence microscopy, however, revealed that tFT-Pex15Δ30 was not detected any more in mitochondria. The GFP signal changed its localization to the cytosol and the ER, whereas mCherry was only detected in the vacuole (*Figure 4B*). This corroborates that tFT-Pex15Δ30 is only transiently localized in the ER and finally gets targeted to the vacuole (where the GFP signal is quenched). Similar observations were made using mutants in other components of the Doa10 ligase complex, *CUE1*, *UBC7* or *UBC6* deletion led to a stabilization of tFT-Pex15Δ30 (*Figure 1C*). Likewise, overexpression of Msp1 in *cue1Δ* or *ubc6Δ* did not restore tFT-Pex15Δ30 levels to wt (*Figure 4A*) and imaging showed relocalization of tFT-Pex15Δ30 from mitochondria to the cytosol, the ER and the vacuole (*Figure 4—figure supplement 1D and E*). These results show that in Doa10-deficient strains Msp1 levels are limiting when expressed from its endogenous promoter and argues that Doa10 defines the predominant route for proteasomal degradation of Pex15Δ30. Consistent with this, we observed also reduced poly-ubiquitination of Pex15Δ30 in the *doa10Δ* strain (*Figure 4C* and *Figure 4—figure supplement 1F*). Interestingly, in the absence of Doa10 a prominent single band resembling mono-ubiquitinated Pex15Δ30 was observed (*Figure 4C*). This indicates that alternative ubiquitination events not leading to proteasomal degradation can occur. The question whether vacuolar degradation compensating ablation of Doa10 depends on ubiquitin-dependent trafficking (*Komander and Rape, 2012*) remains to be addressed.

## Screen for other Msp1 and Doa10 clients

Our data indicates that the truncated Pex15Δ30 reporter is targeted by Msp1- and Doa10-dependent quality control. In order to identify more substrates for this pathway, we measured the abundance of 55 TA proteins (*Burri and Lithgow, 2004*, see *Supplementary file 2*) in wt, *msp1Δ* and *doa10Δ* strains. The corresponding sfGFP-tagged alleles expressed from the constitutive *NOP1* promoter were obtained from an N-terminal library (*Yofe et al., 2016*) and fluorescence was analyzed from colonies. This screen identified six proteins with a more than 2-fold increase in their GFP level upon deletion of *DOA10* (*Figure 5A*). One of them, Ubc6, had been shown to be a substrate of Doa10 in the context of ERAD (*Figure 5A and B*) (*Walter et al., 2001*). Other proteins that accumulated upon compromised function of Doa10 are Csm4, Pgc1, Sps2, Ydl241w. These proteins localized to the ER or other vesicular structures (*Figure 5B*) and did not accumulate in the *msp1Δ* mutant (*Figure 5A*). The strongest stabilization in *doa10Δ* was observed for Fmp32, which is an uncharacterized mitochondrial protein implicated in maintenance of the respiratory capacity of mitochondria (*Paupe et al., 2015*) (*Figure 5A and B*).

A similar analysis in *msp1Δ* identified weak, but significant accumulation of 3 proteins: Pex15, Gos1 and Fmp32 (*Figure 5A*). This indicates that consistent with previous reports (*Chen et al., 2014*), a small fraction of full-length Pex15, as well as Gos1, mislocalized to mitochondria and are subject to Msp1-dependent extraction. In contrast to Pex15Δ30, full-length Pex15 and Gos1 do not accumulate in *DOA10* deletion strains, indicating that in presence of active Mps1 they are efficiently targeted to their native destination. Again, the strongest effect of *MSP1* deletion was observed for Fmp32 (1.8-fold increase) (*Figure 5A*). This indicates that like the Pex15Δ30 reporter, Fmp32 is extracted and degraded in a concerted action by Doa10 and Msp1. Interestingly, under its native promoter, sfGFP-tagged Fmp32 is almost undetectable in wt strains (*Figure 5C and D*). Deletion of *MSP1* led to a robust increase in the levels of sfGFP-Fmp32 (*Figure 5D*). Of note, sfGFP-Fmp32 did not accumulate in the absence of Cis1, indicating that is extracted by the canonical Msp1 activity and not target for mitochondrial protein import stress response (*Weidberg and Amon, 2018*) (*Figure 5—figure supplement 1*). Deletion of *DOA10* alone or in combination with *MSP1* led to a dramatic increase of Fmp32 localizing both to the cytosol and mitochondria (*Figure 5C and D*). This result implies that the mitochondrial protein Fmp32 is constantly degraded by Doa10 with the assistance of Msp1 extracting it from the OMM. Whether this pre-insertion degradation of Fmp32 by Doa10 is important for the abundance control of Fmp32 and whether under certain conditions a stable mitochondrial pool exists, remain important open questions.

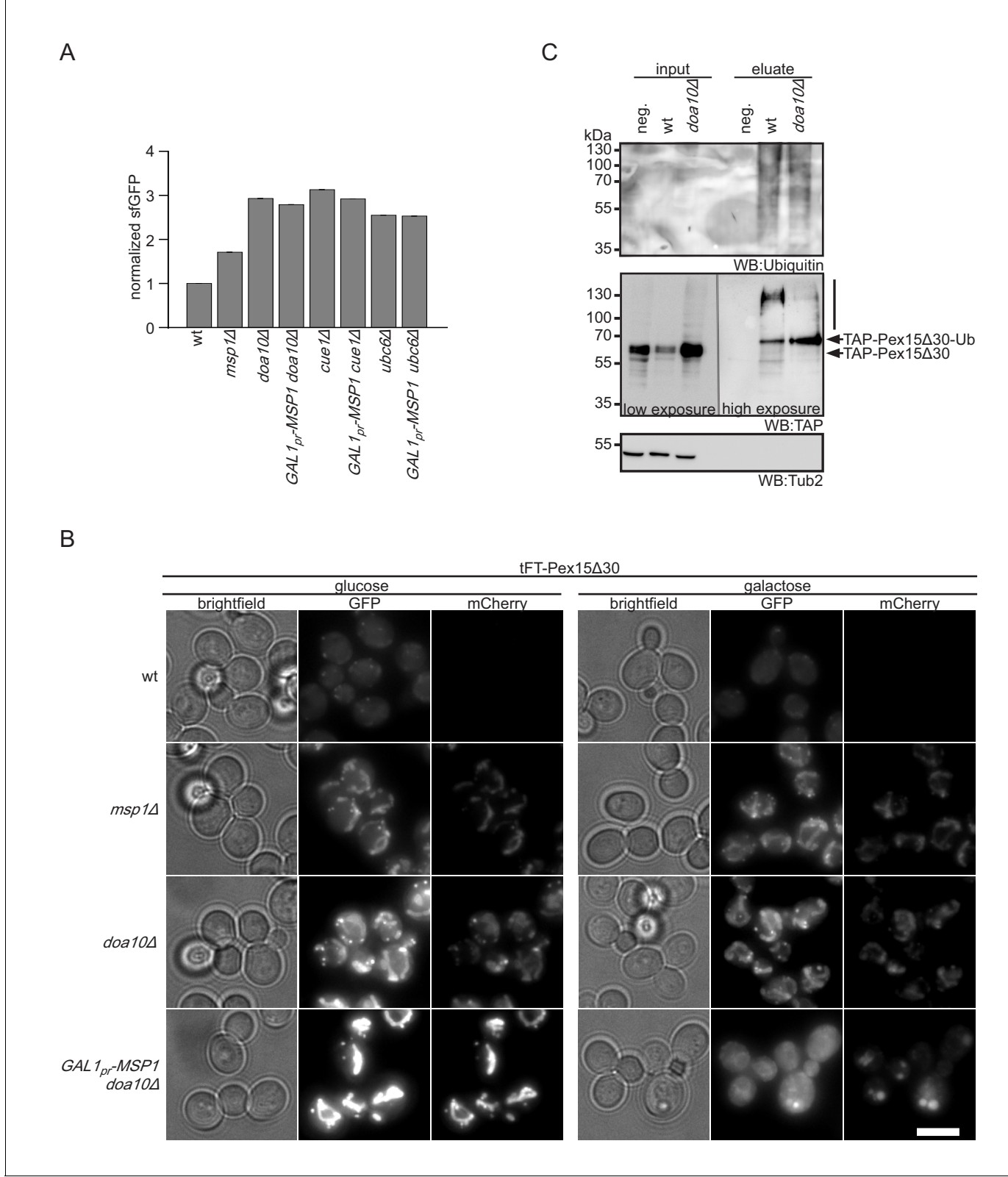

**Figure 4.** Msp1 overexpression clears mitochondrial accumulated Pex15Δ30 in *doa10Δ*. (**A**) Flow cytometry GFP measurements of tFT-Pex15Δ30 expressed in wt, *DOA10*, *CUE1*, *UBC6* and *MSP1* mutant yeast (n = 4, ± SEM). Msp1 protein expression is controlled from the galactose-inducible *GAL1 promoter* (*GAL1pr-MSP1*). (**B**) Localization of tFT-Pex15Δ30 in strains of (**A**) with and without Msp1 overexpression. Scale bar: 5 μm. (**C**) Ubiquitylation of *Figure 4 continued on next page*

*Figure 4 continued*

TAP-Pex15Δ30 in wt and *doa10Δ* strains expressing 10xhistidine-tagged ubiquitin as assessed by Ni-NTA affinity purification and western blotting (WB). neg. refers to a strain expressing TAP-Pex15Δ30 but not 10xhistidine-tagged ubiquitin. Tub2 is used as loading control.

DOI: https://doi.org/10.7554/eLife.45506.008

The following figure supplement is available for figure 4:

**Figure supplement 1.** Overexpression of Msp1 restores cellular Pex15Δ30 level.

DOI: https://doi.org/10.7554/eLife.45506.009

## Oligomerization in the OMM prevents extraction by Msp1 and recognition by Doa10

In vitro studies demonstrated that Msp1 and ATP are necessary and sufficient for the extraction of TA proteins from proteoliposomes (*Wohlever et al., 2017*). Together with our screening data, which did not identify additional non-essential mitochondrial factors needed for Pex15Δ30 turnover, this suggests that Msp1 acts on its clients without co-factors. This raises the question of how Msp1 recognizes its substrates in the OMM membrane. Recent analysis suggested the involvement of a juxta-membrane cytoplasmic hydrophobic patch in Pex15 is important for Msp1-mediated extraction (*Li et al., 2019*). However, Fmp32 and Gem1 do not show this feature (*Figure 6—figure supplement 1A*). Since known Msp1 clients lack any obvious similarity in their TM anchors (*Figure 6—figure supplement 1B*) (*Chen et al., 2014*; *Li et al., 2019*; *Okreglak and Walter, 2014*), we asked whether sequence-independent parameters may determine specificity. One possibility could be that Msp1 extracts proteins that are not stably anchored in the membrane. If this is the case, strengthening OMM association should prevent extraction of such TA proteins. One way to strengthen membrane interaction of proteins is to increase the number of membrane interaction sites per molecule, that is by fusion of a stronger TM anchor or an additional membrane interaction domain. As this is not possible for TA proteins, we reverted to regulated protein dimerization, which increases avidity to the membrane. To this end we utilized the rapamycin-inducible protein-protein interaction of the FRB1 and FKBP12 protein domains (*Choi et al., 1996*). We fused the TM domain of Pex15 to GFP-FRB1 or HA-FKBP12 (further on called FRB1$^{TMD}$ and FKBP12$^{TMD}$, respectively) and investigated their behavior upon rapamycin-induced dimerization (*Figure 6A*). The FRB1$^{TMD}$ protein localized to mitochondria (*Figure 6B*). In order to confirm mitochondrial localization, the dimerization partner FKBP12$^{TMD}$ was fused to a mCherry fluorophore (mChe-FKBP12$^{TMD}$) (*Figure 6—figure supplement 1C*). Whereas FRB1$^{TMD}$ fusion protein showed a half-life in the range of the Pex15Δ30 reporter (*Figure 3C*, *Figure 6C* and *Figure 6—figure supplement 1D*), FKBP12-Pex15$^{TMD}$ was more stable (*Figure 6—figure supplement 1E*). Since similar stabilizing effect of FKBP12 fusion proteins have been reported previously (*Edwards and Wandless, 2007*; *Morgan et al., 2014*), we restricted our analysis to the FRB1$^{TMD}$ reporter and used the FKBP12 constructs as mimic for a stable complex partner. Importantly, the FRB1$^{TMD}$ reporter was degraded in an Msp1- and Doa10-dependent manner (*Figure 6—figure supplement 1F*). Addition of rapamycin completely stabilized FRB1$^{TMD}$ when co-expressed with FKBP12$^{TMD}$ (*Figure 6B,C* and *Figure 6—figure supplement 1D*) whereas no effect was observed in strains only expressing FRB1$^{TMD}$ (*Figure 6—figure supplement 1G*). Similar results were observed with a larger mChe-FKBP12$^{TMD}$ fusion partner (*Figure 6—figure supplement 1D*). In contrast, co-expression of soluble cytoplasmic FKBP12 constructs slightly increased half-life of FRB1$^{TMD}$ from 12 min to 20 min (*Figure 6D* and *Figure 6—figure supplement 1H*). Taken together these results showed that dimerization with a membrane integral protein counteracts Msp1 extraction and subsequent degradation of our Pex15-based reporter. The partial stabilization of FRB1$^{TMD}$ in complex with soluble FKBP12 constructs suggests that the Msp1-mediated extraction is influenced by energy required for unfolding and disassembly of the cytoplasmic client domain. In conclusion, the result of the induced-dimerization experiment supports the model that Msp1 acts as unspecific extraction factor for monomeric TA proteins and, as observed in other cellular organelles, protein maturation is accompanied by formation of stable multiprotein complexes (*Juszkiewicz and Hegde, 2018*).

If the oligomeric state is the main factor driving Msp1 recognition, we next asked whether Msp1 also acts on subunits of native OMM complexes. An example is Gem1, which is a TA protein associated with the ER-mitochondria-tethering complex ERMES (ER-mitochondria encounter structure)

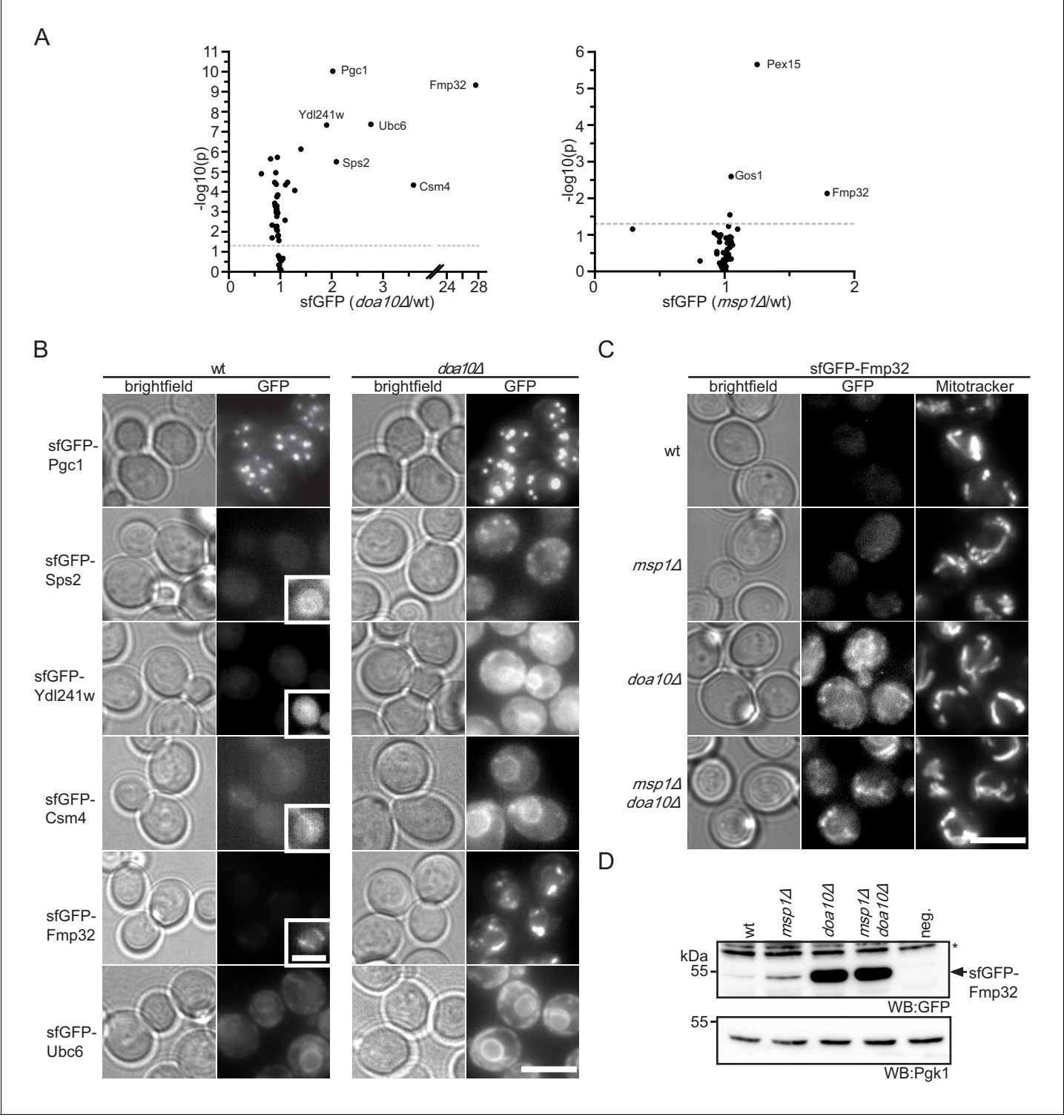

**Figure 5.** Doa10 controls targeting fidelity of mitochondrial TA proteins. (**A**) Mean GFP fluorescence of 55 N-terminally GFP-tagged TA proteins expressed from *NOP1* promoter deleted for *DOA10* or *MSP1* compared to wt. Measurements were taken from colonies grown on agar (n = 4). Highlighted proteins are significantly enriched (Student's t-test p<0.05, threshold gray dashed line). (**B**) Microscopy validation of significantly enriched proteins from (**A**) in wt compared to *doa10Δ*. Images for each protein investigated are shown with the same display range for wt and *doa10Δ*. Insets show optimized display range. Scale bar: 5 µm. (**C**) Microscopy analysis of sfGFP-tagged Fmp32 under its endogenous promoter. Mitochondria are stained with the mitochondrial dye Mitotracker Red. Scale bar, 5 µm. (**D**) Western blot (WB) analysis of log phase growing yeast from (**C**). neg. is a wt yeast referring to GFP specific bands. Unspecific bands are marked with star. Pgk1 is used as loading control.

*Figure 5 continued on next page*

*Figure 5 continued*

DOI: https://doi.org/10.7554/eLife.45506.010

The following figure supplement is available for figure 5:

**Figure supplement 1.** Msp1 does not require Cis1 for Fmp32 turnover.

DOI: https://doi.org/10.7554/eLife.45506.011

(*Kornmann et al., 2011*) that was not included in our TA array (*Supplementary file 2*). We expressed sfGFP-Gem1 from the *NOP1* promoter. Consistent with our hypothesis, a weak but significant increase in sfGFP-Gem1 levels were detected in the *msp1Δ* strains (*Figure 6E*). As observed for the Pex15Δ30 reporter and Fmp32, deletion of *DOA10* and the *msp1Δdoa10Δ* mutant increased sfGFP-Gem1 levels. This shows, that Gem1 is an additional client for both, Msp1 and Doa10. Taken together with previous observation that Gem1 is only stable in presence of the other ERMES subunits (*Kornmann et al., 2011*), this result implies that complex formation is a primary determinant preventing extraction by Msp1.

## Discussion

The AAA-ATPase Msp1 extracts mislocalized TA proteins from the OMM and thus comprises an important proteostasis safeguard of the cell (*Chen et al., 2014*; *Okreglak and Walter, 2014*). We herein characterized the fate of TA proteins after they have been extracted from the OMM. A genome-wide screen revealed that the ER-resident Doa10 E3 ubiquitin ligase complex is responsible for targeting these TA proteins for proteasomal degradation. Since, we did not identify any additional factor, we suggest that the Msp1 dislocase does not need any assistance for substrate recognition and extraction. We could also show that the oligomeric state of TA proteins in the OMM regulates Msp1-mediated extraction. Furthermore, our substrate screen enlarges the set of known Msp1 clients by Fmp32 and Gem1 corroborating that a heterogenous set of clients are recognized. Likewise, these additional substrates do not show a hydrophobic juxtamembrane patch that recently has been suggested to be crucial for recognition of TA proteins by Msp1 (*Li et al., 2019*). Based on these results, we suggest that productive localization of mitochondrial TA proteins is governed by an equilibrium between complex assembly and extraction of orphan subunits whereas Doa10 determines the cytoplasmic concentration of targeting competent species (*Figure 7*).

### Doa10 controls TA abundance and protein targeting fidelity

Although Doa10 is a central player of ERAD (*Mehrtash and Hochstrasser, 2018*; *Ruggiano et al., 2014*), initially it has been identified in a screen for mutants with a defect in turnover of the soluble nuclear transcription factor MATα2 (*Swanson et al., 2001*). Consistent with this function in recognizing soluble proteins, it has been also linked to a pre-insertion ER quality control pathway of GPI-anchored proteins (*Ast et al., 2014*), which are commonly targeted to the Sec61 translocon post-translationally by an N-terminal SRP-independent signal sequence. Similarly, TA proteins are post-translationally inserted into the ER, making them susceptible for premature aggregation or degradation. However, targeting factors such as Get3 and general chaperones keep them in a transport and insertion competent state thereby preventing protein aggregation (*Cho and Shan, 2018*; *Schuldiner et al., 2008*; *Voth et al., 2014*). Here, we now show that Doa10 samples and targets certain TA proteins prior their membrane insertion for proteasomal degradation distinct of its ERAD function. However, certain key ERAD factors such as Cdc48 and Ubx2 were not analyzed in our screens because they were not present in our library (*Li et al., 2011*; *Winzeler et al., 1999*). While in the artificial Pex15-derived reporter (Pex15Δ30) we assume that the TM domain adopts a non-native conformation, we also show that the endogenous mitochondrial proteins Fmp32 and Gem1 are Doa10 substrates. This suggests, that Doa10 controls the cytosolic level of these proteins and by this regulates their OMM insertion. Therefore, we propose that targeting fidelity of TA proteins to cellular membranes is ensured by Doa10-dependent abundance control (*Figure 7*). However, this does not apply to all TA proteins. The full-length Pex15 protein only accumulates in the OMM upon compromised Msp1 function. Doa10 deletion does not lead to a significant increase in cellular Pex15 level, indicating that it is a better client for the GET pathway and/or other peroxisomal targeting

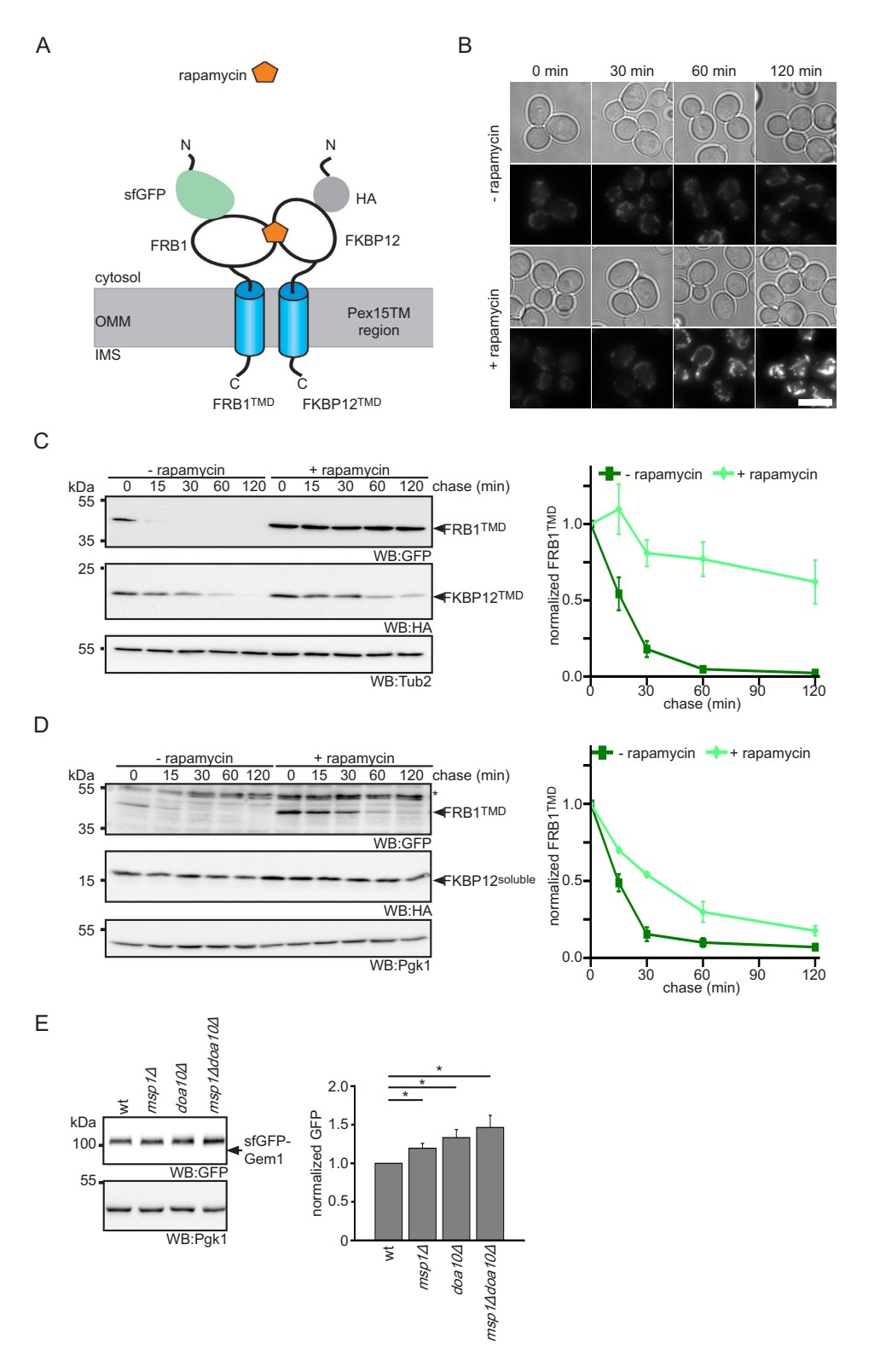

**Figure 6.** Protein dimerization impedes Msp1-dependent extraction. (**A**) Scheme of rapamycin-induced dimerization for the two reporter proteins FRB1TMD and FKBP12TMD. (**B**) Microscopy analysis of wt yeast expressing FRB1TMD and FKBP12TMD. Images were taken with and without rapamycin treatment for the indicated time. Scale bar: 5 μm. (**C**) Cycloheximide chase of strains from (**B**) with and without rapamycin pre-treatment for 30 min. Quantification of FRB1TMD (n = 3, ± SEM) normalized to t = 0 of untreated sample. WB, western blot. Tub2 is used as loading control. (**D**)

*Figure 6 continued on next page*

*Figure 6 continued*

Cycloheximide chase of strains expressing FRB1$^{TMD}$ and cytosolic FKBP12 (FKBP12$^{soluble}$) with and without rapamycin pre-treatment for 30 min. Quantification of FRB1$^{TMD}$ (n = 3, ± SEM) normalized to t = 0 of untreated sample. Unspecific bands are marked with star. Pgk1 is used as loading control. (**E**) Steady state analysis of sfGFP-Gem1 expressed from the *NOP1* promoter in wt, *msp1Δ*, *doa10Δ* and *msp1Δdoa10Δ* with quantification (n = 6 ± SEM, star indicates Student's t-test p<0.05).

DOI: https://doi.org/10.7554/eLife.45506.012

The following figure supplement is available for figure 6:

**Figure supplement 1.** Enhanced membrane association impedes Msp1-dependent extraction of Pex15-derived reporters from mitochondria.
DOI: https://doi.org/10.7554/eLife.45506.013

routes. Other TA proteins reported to be extracted by Msp1 from the OMM in a *GET3* mutant are Gos1 (*Chen et al., 2014*) Frt1 and Ysy6 (*Li et al., 2019*), which we do not see accumulating to the same extent. This likely results from different expression levels and the complex genetic interaction between *MSP1* and the GET pathway. In contrast, our microscopy analysis revealed substantial mitochondrial accumulation of Fmp32 in *msp1Δ* and *doa10Δ* strains. Not much is known about the physiological function of Fmp32. It has been implicated in calcium storage and mutation causes a phenotype similar to cytochrome c assembly defect (*Paupe et al., 2015*).

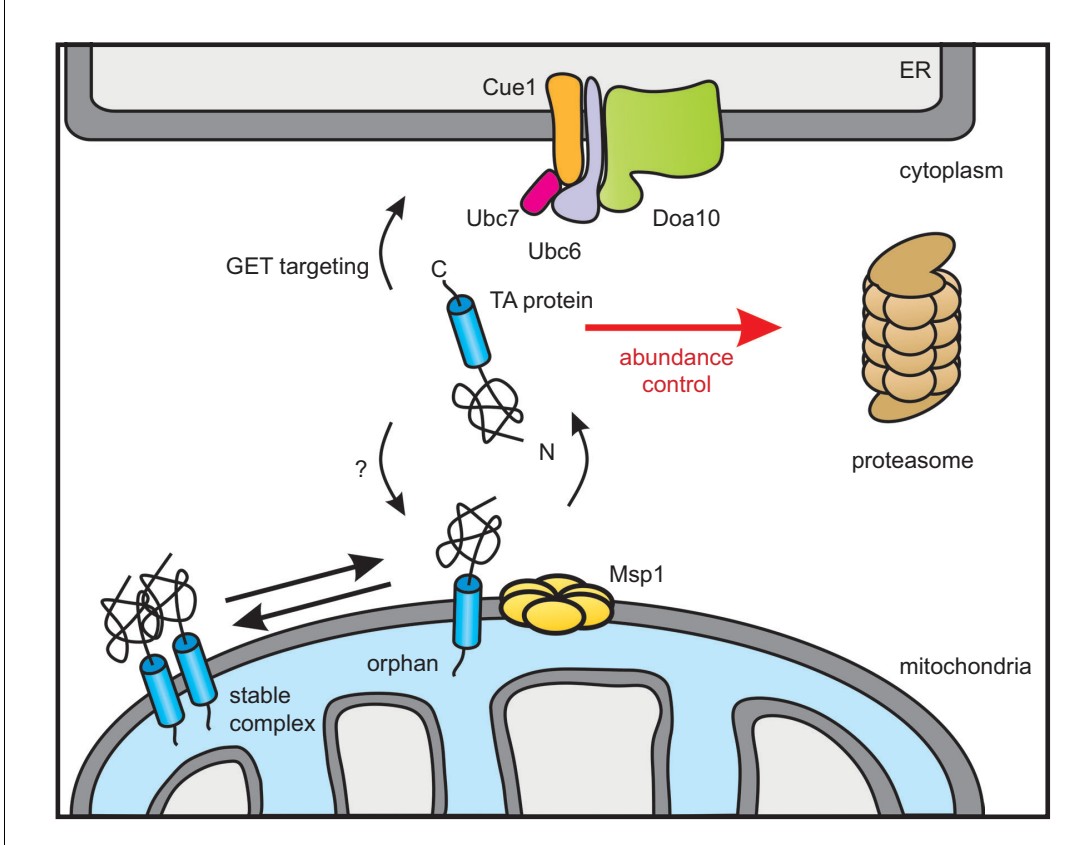

**Figure 7.** Model of Doa10-mediated TA protein abundance control increasing targeting fidelity and removing clients of Msp1 dislocase. TA proteins are post-translationally targeted to the ER by the GET pathway or insert into the OMM by an so far unknown mechanism. In the OMM, TA proteins become subject for Msp1-mediated extraction unless they dimerize or form hetero-oligomers (not shown). The ER-resident E3 ubiquitin ligase Doa10 together with its complex partners targets surplus TA proteins including Msp1-clients from the cytosol for proteasomal degradation in order to improve targeting fidelity and abundance control.
DOI: https://doi.org/10.7554/eLife.45506.014

# Msp1 extends OMM surveillance mechanisms for extraction of monomeric TA proteins

Mitochondria possess a multilayered protein quality control system that is essential to maintain the integrity of mitochondrial functionality (*Moehle et al., 2019*). At the inner mitochondrial membrane, two AAA protease complexes referred to as *m*- and *i*-AAA proteases select damaged proteins for degradation in the matrix and intermembrane space, respectively (*Patron et al., 2018*). In addition to dealing with proteins of the inner membrane, the active subunit of the *i*-AAA protease Yme1 has also been shown to recognize OMM proteins such as Tom22 and Om45 if they accidentally appear at the inner mitochondrial side (*Wu et al., 2018*). At the OMM, the cytoplasmic AAA-ATPase Cdc48 extracts ubiquitinated protein for proteasomal degradation in a process that is regulated by the substrate-processing factor Doa1 (*Wu et al., 2016*) and Ubx2 (*Mårtensson et al., 2019*). Given these powerful proteostasis factors, why do eukaryotic cells have an additional Msp1-dependent surveillance mechanism? One reason probably is that both Yme1 and Cdc48-mediated degradation require recognition of aberrant protein domains exposed into the intermembrane space or into the cytosol, respectively. Msp1 completes surveillance of the OMM by recognizing TA proteins that do not possess any large C-terminal regions protruding into the intermembrane space nor contain any obvious folding defect in the cytoplasm.

Since our genetic screen of all non-essential genes in yeast did not identify any additional factor for recognition and extraction of TA proteins from the OMM, we tested the hypothesis that only monomeric TA clients can be extracted by Msp1. Such clients are likely to be proteins that insert into the OMM unassisted of targeting factors in which case Msp1 functions to shift the dynamic equilibrium towards a localization of these proteins in the cytoplasm. A live cell quantitative microscopy approach combined with a computational analysis of Msp1 in peroxisomes suggested that interaction of Pex15 with the peroxisomal protein Pex3 is the determinant that prevents Msp1-mediated extraction (*Weir et al., 2017*). By applying chemical-induced dimerization to a mitochondria-localized Pex15 TM domain reporter we could show that only monomeric forms can be extracted by Msp1 even though both Doa10-mediated degradation and Msp1 are functional. Likewise, we observe that the overexpressed TA protein Gem1 becomes a Msp1 client when its native complex partners are limiting. Thus, we propose that Msp1 samples the OMM and extracts monomeric TA proteins, whereas complex assembly counteracts this reaction. Extraction may either be prevented by a steric clash of bulky cytoplasmic/IMS domains of the dimeric client protein or by the tandem TM segments exceeding the pulling force of the Msp1 dislocase. Interestingly, a recent biochemical and structural analysis revealed that Msp1 acts as a membrane-anchored ATP-dependent ring hexamer with client proteins likely being extracted through the pore (*Wohlever et al., 2017*). The Msp1 TM anchor and N-terminal linker domain, however, are dispensable for the in vitro dislocation reaction (*Wohlever et al., 2017*). Consistent with the idea that Msp1 does not sample TM domains of clients, we and others revealed that Msp1 extracts a heterogenous set of TA proteins lacking any obvious similarity in their TM anchors (*Figure 6—figure supplement 1B*) (*Chen et al., 2014*; *Li et al., 2019*; *Okreglak and Walter, 2014*). A juxtamembrane hydrophobic patch in Pex15Δ30 has been shown to facilitate recruitment to Msp1 (*Li et al., 2019*), but our analysis of Fmp32 and Gem1 show that this feature is not diagnostic (*Figure 6—figure supplement 1A*). However, the extent of OMM anchoring and/or complex assembly are important parameters. These principles possibly also apply to the recently described function in resolving clogged TOM complexes (*Weidberg and Amon, 2018*). Even though Msp1 is recruited to the TOM complex by the cytosolic protein Cis1, extraction of the incompletely inserted import intermediate is possible because the protein is not associated with complex partners. Of note, our genetic screen and a recent study (*Li et al., 2019*) imply that Cis1 is not required for the extraction of OMM TA proteins, revealing that Msp1 is a versatile dislocase that is used for multiple purposes.

In conclusion, our results reveal a link between the ER-resident E3 ubiquitin ligase Doa10 and the OMM proteostasis control. Taken together with a recent report on an ER surface retrieval pathway for inner mitochondrial membrane proteins by the ER-localized chaperone Djp1 (*Hansen et al., 2018*), our study reveals a so far unanticipated close interplay between these two organelles controlling protein abundance and targeting fidelity. More generally, control of complex assembly by potent mechanisms removing orphan subunits emerges as a widely used principle (*Juszkiewicz and Hegde, 2018*). However, while degradation of unassembled subunits in the cytoplasm and the ER is

directly linked to E3 ubiquitin ligases, Msp1 appears to act relatively unspecific. In a separate process, Doa10 samples insertion competent TA proteins to ensure their targeting fidelity, possibly preventing endless futile cycles of insertion and extraction. Further studies are needed to define the exact molecular determinants that are recognized by Msp1 and Doa10. Homologues for Msp1 (ATAD1/Thorase) and Doa10 (MARCH6/TEB4) exist in humans, indicating a conserved role of both proteostasis factors in the abundance and quality control of mitochondrial TA proteins.

## Materials and methods

### Yeast strains and plasmid construction

All yeast strains used in this study are based on BY4741, BY4742 or Y8205. Strain construction was performed according to standard protocols for chromosomal yeast manipulation (*Janke et al., 2004*; *Knop et al., 1999 Khmelinskii et al., 2011*). Strain validation was performed by colony PCR. Strains are listed in *Supplementary file 3*. If not otherwise mentioned all yeast strains were cultured in YPD medium (10 g/l Bacto yeast extract, 20 g/l Bacto peptone) supplemented with 2% (w/v) glucose or 2% (w/v) raffinose and 2% (w/v) galactose or defined synthetic complete (SC) medium (6.7 g/l Bacto yeast nitrogen base without amino acids, 2 g/l amino acid dropout mix, 2% (w/v) carbon source) at 30°C, shaking.

For screening, the non-essential subset of the heterozygous diploid yeast deletion collection was used comprising 4642 mutants (*Winzeler et al., 1999*). The TA array according to *Burri and Lithgow (2004)* was compiled from N-terminally sfGFP-tagged strains obtained from the N-SWAT library (*Yofe et al., 2016*). TA proteins investigated are listed in *Supplementary file 2*.

Plasmids used in this context were constructed using standard cloning procedures and verified by sequencing (Eurofins). All plasmids are listed in *Supplementary file 4*.

If not otherwise noted the standard sfGFP-mCherry timer fusion was used for all tFT-tagged proteins (*Khmelinskii and Knop, 2014*). For colocalization and other studies with GFP, a circular permuted variant of sfGFP (sfGFPcp8) was used (*Khmelinskii et al., 2016*). To generate the reporter construct, the open reading frame of Pex15Δ30 (*Okreglak and Walter, 2014*) was cloned into yeast vectors and fused to a N-terminal tFT-tag (*Khmelinskii et al., 2012*). tFT-tag was exchanged for sfGFPcp8 or TAP-tag in order to obtain other Pex15Δ30 fusions used throughout the study. The FRB1[TMD] dimerization constructs were created by fusing the FKBP12 rapamycin binding protein (FRB1) to the C-terminal sequence of Pex15 (from amino acid position 285 to amino acid position 353). To follow cellular distribution this fusion construct was N-terminally fused with sfGFPcp8 via a KLGAGA linker. The FKBP12[TMD] dimerization partner including the FK506 binding protein (FKBP12) was similarly designed. FKBP12, with a single HA tag at its N-terminus was fused to the Pex15 C-terminus comprising amino acid position 285 to amino acid position 353. The second membrane anchored mCherry-FKBP12[TMD] construct was cloned by extending the N-terminal region with a HA-mCherry and a GTSAGAGAGAGA linker. Soluble FKBP12 dimerization partners were created by deleting the Pex15-derived membrane anchor.

### Antibodies

For immunoblot detection the following primary antibodies were used: rabbit anti-GFP polyclonal antibody (Abcam, ab6556), mouse anti-ubiquitin (BioLegend), rabbit PAP (*Khmelinskii et al., 2014*), mouse anti-Pgk1 monoclonal antibody (Molecular Probes, 22C5D8) or Tub2 (gift from Elmar Schiebel, ZMBH, Heidelberg University) as loading control. For secondary antibody detection HRP-conjugates were used: Donkey IgG anti-Mouse IgG (Dianova, Jackson ImmunoResearch, 715-035-150), Donkey IgG anti-Rabbit IgG (Dianova, Jackson ImmunoResearch, 711-035-152).

### Synthetic genetic arrays and screening

Automated mating, sporulation and haploid selection of arrayed yeast colonies was performed by sequential pinning on appropriate selective media using a RoToR pinning robot (Singer Instruments, UK) as previously described (*Baryshnikova et al., 2010*).

For the genome-wide screen, query strains AK1306 and AK1307 were crossed with a heterozygous yeast deletion collection (*Winzeler et al., 1999*) in 1536-colony format with four technical replicates of each cross. Technical replicates were arranged next to each other. Fluorescence intensities

of the final colonies were measured after 22 hr of growth on synthetic complete medium lacking leucine and supplemented with adenine (200 mg/l) at 30°C using an Infinite M1000 Pro plate reader equipped with stackers for automated plate loading (Tecan) and custom temperature control chamber. Measurements in mCherry (587/10 nm excitation, 610/10 nm emission, optimal detector gain) and sfGFP (488/10 nm excitation, 510/10 nm emission, optimal detector gain) channels were performed at 400 Hz frequency of the flash lamp, with ten flashes averaged for each measurement. Measurements were filtered for potentially failed crosses based on colony size. Fluorescence intensity measurements were log-transformed and the data was normalized for spatial effects on plates by local regression. To estimate the changes in protein abundance (sfGFP intensity) and stability (mCherry/sfGFP ratio), z-scores were computed by subtracting the median across plates and dividing by the median absolute deviation. Technical replicates were then summarized by calculating the mean and standard deviation. P-values were computed using a t-test and adjusted for multiple testing by controlling the false discovery rate using the method of Benjamini-Hochberg.

For the targeted screens, query strains AK1305, AK1306 and AK1307 were crossed with an array of haploid strains carrying mutations in components of the ubiquitin-proteasome system. All mutants – gene deletions of non-essential genes (*Winzeler et al., 1999*) and temperature-sensitive alleles of essential genes (*Li et al., 2011*) – were validated by colony PCR to verify the genomic location of the kanMX selection marker. Fluorescence intensities of the final colonies were measured after 36 hr of growth on synthetic complete medium lacking histidine and supplemented with adenine (200 mg/l) at 30°C using a Spark plate reader (Tecan). Measurements were filtered for potentially failed crosses based on colony size. Fluorescence intensity measurements were corrected for background fluorescence locally and normalized to the screen median.

For the TA array screen query strains YVD247, YVD244 or YVD238 were crossed with N-SWAT GFP-tagged TA fusions (*Yofe et al., 2016*) in 384-colony format. Non-fluorescent control colonies were arranged next to the strains of interest. Whole colony fluorescence measurements were performed after 20 hr of growth on synthetic complete medium supplemented with G418. Fluorescence intensity measurements were corrected for local background fluorescence. Mean values were calculated from four replicates and standard deviation was calculated. p-values were determined using Student's t-test.

## Preparation of cell extracts and western blot analysis

Total cell extracts were prepared using alkaline lysis followed precipitation with trichloroacetic acid (TCA) (*Knop et al., 1999*). Proteins were dissolved in SDS sample buffer (50 mM Tris-Cl pH 6.8, 10 mM EDTA, 5% glycerol, 2% SDS, 0.01% bromphenol blue) and analyzed by SDS-PAGE. Semi-dry western blotting was used to transfer proteins onto PVDF membranes (Merck Millipore). Immunodetection was performed using appropriate antibodies. Immunoreactive species were visualized using WesternBright ECL (Biozym) and detected using the LAS-4000 Fuji Imager.

## Steady-state and cycloheximide chase experiments

For steady state analysis saturated grown yeast cultures were diluted and allowed to reach log phase by at least two doublings. Then 1 ml sample was harvested by centrifugation and cell extracts were prepared using alkaline lysis. For cycloheximide chase experiments 0.1 mg/ml cycloheximide was added to log phase growing yeast cultures. 1 ml aliquots were harvested at each time point by centrifugation for cells grown in YPD. If cells were grown in SC medium, 1 ml sample was directly mixed with β-mercaptoethanol/NaOH and flash frozen prior to TCA extraction. Lysates were analyzed by SDS-PAGE and western blotting. Data shown are representative of three independent experiments.

## Proteasome inhibition experiments

To inhibit the proteasome *pdr5Δ* yeast strains were created and treated with 80 µM MG132 and 80 µM bortezomib for 45 min before cells were harvested or cycloheximide chase was performed.

## Ubiquitination assay

Ubiquitination status of Pex15Δ30 was determined adapting a protocol from *Khmelinskii et al. (2014)*. Cells co-expressing TAP-tagged Pex15Δ30 and 10xhistidine-tagged ubiquitin from plasmids were cultured in SC-medium lacking histidine and leucine to keep selection pressure. Approximately

$1 \times 10^9$ log phase growing cells were treated for 10 min with 10 mM N-ethylmaleimide (NEM) and 1 mM phenylmethylsulfonylfluorid, harvested, washed with water, resuspended in 1.85 mM NaOH with 15% β-mercaptoethanol and flash frozen in liquid nitrogen. After alkaline lysis proteins were precipitated with a final concentration of 5% TCA. Proteins were pelleted by 10 min centrifugation at 20,000xg and washed with ice cold acetone. Pellet was resuspended in 3 ml guanidium buffer (6 M guanidinium chloride, 100 mM Tris-Cl pH 9.0, 300 mM NaCl, 10 mM imidazole, 0.2% Triton X-100 and 5 mM chloroacetamide). After solubilization, suspension was clarified for 30 min centrifugation with 20,000xg. Sample was incubated with TALON Metal Affinity Resin (Clontech) for 90 min rotating at room temperature. Beads were washed twice with wash buffer (8 M urea, 100 mM sodium phosphate pH 7.0, 300 mM NaCl, 5 mM imidazole, 0.2% Triton X-100 and 5 mM chloroacetamide) and twice with wash buffer supplemented with 0.2% SDS. Finally, ubiquitin conjugates were eluted with 100 µl elution buffer (8 M urea, 100 mM sodium phosphate pH 6.5, 300 mM NaCl, 250 mM imidazole, 0.2% Triton X-100, 0.2% SDS and 5 mM chloroacetamide). For input 0.3% of the protein amount used for purification was analyzed with eluate by SDS-PAGE followed by western blot analysis.

## Immunoprecipitation

$1 \times 10^9$ cells expressing tFT-Pex15Δ30 were treated with 80 µM MG135 and 100 µM bortezomib for 1.5 hr before harvesting by centrifugation and washing with water. Cells were resuspended in lysis buffer (50 mM Tris-Cl pH7.5, 150 mM NaCl, 5 mM EDTA, 5 mM NEM, protease inhibitors, 0.5 mM dithiothreitol) and lysed using glass beads. Cell debris were pelleted by 5 min centrifugation with 1200xg. Proteins were solubilized supplementing the suspension with 1% Triton X-100 and incubation on ice for 15 min. After clarification solubilized proteins were incubated with 20 µl GFP specific single chain antibody fragment coupled to sepharose beads (*Fleig et al., 2012*; *Rothbauer et al., 2008*) for 1 hr and incubated at 4°C. Antibody fragment coupled beads were washed three times with lysis buffer supplemented with 0.5% Triton X-100. Protein elution was performed by adding 2x SDS sample buffer and heating for 15 min at 65°C.

## Growth assay

Growth curves were acquired from yeast strains grown in liquid medium. Overnight cultures were diluted to $OD_{600}$ 0.01 in YPD medium in 24-well plates and incubated in a TECAN plate reader shaking with 372 rpm at 30°C. Cell density was measured by absorbance measurements at 600 nm averaged from 25 flashes was measured every 5 min. Measurements were corrected against YPD medium blank.

## Overexpression of Msp1

For Msp1 overexpression studies the endogenous promoter of *MSP1* was replaced with the galactose inducible *GAL1* promoter. Msp1 expression was restricted when strains were cultured in medium supplemented with 2% glucose. Msp1 expression was released by exchanging growth medium supplemented with 2% galactose. Therefore, cells were collected through centrifugation for 5 min with 500xg, washed with sterile water and resuspended and diluted in medium supplemented with 2% (w/v) raffinose and 2% (w/v) galactose. Medium switch was performed 4 hr prior to measurement.

## Chemical induced dimerization

For chemical induced dimerization experiments cells were grown to mid-log phase. Dimerization was induced by adding 2 µg/ml rapamycin (Sigma Aldrich).

## Fluorescence microscopy

For imaging, cells expressing fluorescent proteins were grown to mid-log phase in SC medium (filter sterilized). For staining of mitochondria, cells were treated with the mitochondrial dye MitoTracker Red (ThermoScientific, 1:10 000) for 10 min at 30°C. Cells were washed with water and gently resuspended in growth medium. 0.25 $OD_{600}$ equivalents of cells were transferred into 96-well Concavanalin A (Sigma Aldrich) coated glass bottom plates (MGB096-1-2-LG-L, Matrical) (*Khmelinskii and Knop, 2014*). Images were acquired using a DeltaVision Elite microscopy system (GE Healthcare)

equipped with an LED light source (SpextraX, Lumencor), 475/28 and 575/25 nm excitation, and 525/50 and 624/40 nm emission filters (Semrock), a dual-band beam splitter 89021 (Chroma Technology), using either a 60x NA 1.42 PlanApoN or 100x NA 1.4 UPlanSApo oil immersion objectives (Olympus), an sCMOS camera (pco.edge 4.2, PCO), and a motorized stage contained in a temperature-controlled chamber.

### Image quantification
Western blots and microscopy images were processed and quantified using ImageJ (http://rsb.info.nih.gov/ij/). Microscopy images were subtracted for local background before a maximum intensity projection was applied. If not otherwise mentioned all fluorescence images in one figure are displayed with the same display range. Brightfield images are adjusted for optimal display range.

### Flow cytometry
Flow cytometry assays were performed in 96-well format using BD FACSCanto RUO HTS equipped. Data was processed using FlowJo Software. For measurements, strains were grown to saturation in SC medium with appropriate selection pressure. 100 µl SC medium was inoculated from this saturated preculture using a ROTOR HDA handling robot (Singer Instruments, UK) by pinning 1x with long tips. Cells were grown for 11 hr shaking at 30°C to mid-log phase. Fluorescence intensities of 10.000 events were measured (Detector 488-B with 505LP, 530/30 BP filter used for GFP, 561 C with 600LP and 610/20 BP filter for mCherry) in triplicate. Fluorescence measurements were corrected using the mean of a triplicate measurement from a non-fluorescent control. p-values were calculated using Student's t-test.

## Acknowledgements
We thank Matthias Meurer for help with the synthetic gene array infrastructure, advice and critical reading of the manuscript, the ZMBH Flow Cytometry and FACS Core Facility for assistance, Julia Knopf for critical reading of the manuscript and Peter Walter for advice. This study was supported by the Deutsche Forschungsgemeinschaft (DFG) Collaborative Research Center SFB1036 (AK, MK, MKL) and a Starting Grant from the European Research Council ERC-2017-STG#759427 (AK).

## Additional information

### Funding

| Funder | Grant reference number | Author |
|--------|------------------------|--------|
| Deutsche Forschungsgemeinschaft | SFB1036/2 TP10 | Anton Khmelinskii Michael Knop |
| Deutsche Forschungsgemeinschaft | SFB1036/2 TP12 | Marius K Lemberg |
| European Research Council | ERC-2017-STG#759427 | Anton Khmelinskii |

The funders had no role in study design, data collection and interpretation, or the decision to submit the work for publication.

### Author contributions
Verena Dederer, Conceptualization, Data curation, Formal analysis, Validation, Investigation, Visualization, Methodology, Writing—original draft, Writing—review and editing; Anton Khmelinskii, Conceptualization, Data curation, Formal analysis, Supervision, Funding acquisition, Investigation, Visualization, Methodology, Writing—review and editing; Anna Gesine Huhn, Data curation, Methodology; Voytek Okreglak, Conceptualization, Resources, Writing—review and editing; Michael Knop, Conceptualization, Supervision, Funding acquisition, Methodology, Writing—review and editing; Marius K Lemberg, Conceptualization, Resources, Formal analysis, Supervision, Funding acquisition, Investigation, Visualization, Methodology, Writing—original draft, Project administration, Writing—review and editing

## Author ORCIDs
Anton Khmelinskii (iD) http://orcid.org/0000-0002-0256-5190
Anna Gesine Huhn (iD) http://orcid.org/0000-0003-4798-4951
Marius K Lemberg (iD) https://orcid.org/0000-0002-0996-1268

## Decision letter and Author response
Decision letter https://doi.org/10.7554/eLife.45506.021
Author response https://doi.org/10.7554/eLife.45506.022

## Additional files

### Supplementary files
• Supplementary file 1. sfGFP and mCherry/sfGFP ratio z-scores measured for tFT-Pex15Δ30 and tFT-Tom5 in the non-essential yeast deletion collection.
DOI: https://doi.org/10.7554/eLife.45506.015

• Supplementary file 2. Array of investigated TA proteins according to *Burri and Lithgow (2004)*.
DOI: https://doi.org/10.7554/eLife.45506.016

• Supplementary file 3. Yeast strains used in this study.
DOI: https://doi.org/10.7554/eLife.45506.017

• Supplementary file 4. Plasmids used in this study.
DOI: https://doi.org/10.7554/eLife.45506.018

• Transparent reporting form
DOI: https://doi.org/10.7554/eLife.45506.019

### Data availability
All data generated or analysed during this study are included in the manuscript and supporting files.

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
