## [Decision Letter]

Thank you for submitting your article "Cooperation of mitochondrial and ER factors in quality control of tail-anchored proteins" for consideration by *eLife*. Your article has been reviewed by three peer reviewers, one of whom is a member of our Board of Reviewing Editors, and the evaluation has been overseen by Anna Akhmanova as the Senior Editor. The reviewers have opted to remain anonymous.

The reviewers have discussed the reviews with one another and the Reviewing Editor has drafted this decision to help you prepare a revised submission.

Summary:

How cells deal with mis-targeted and mis-localized proteins is an important question. Tail-anchored proteins that mis-target to mitochondria and peroxisomes have been shown in earlier studies to be extracted by the AAA-ATPase Msp1. It has been suggested for one model substrate that Msp1 selects substrates that have failed to assemble with partners. Whether this principle is more broadly applicable and how Msp1 substrates are degraded has not been clear.

The first, and primary, advance of this study is a systematic analysis in yeast that identifies the ubiquitin ligase Doa10 as the factor required for degradation within the Msp1 pathway. It is shown to act after Msp1-mediated extraction. Together with the previously established (and confirmed here) idea that Doa10 also targets proteins that fail targeting, it seems to be a key factor limiting the amount of free untargeted TA proteins in the cytosol. In the absence of Doa10, this study suggests that TA proteins are instead trafficked from the ER to vacuole as an alternative degradation route. The second advance is the extension of an earlier idea about Msp1 substrates being unassembled TA proteins. The third advance in the study is to identify native Msp1-pathway substrates, the most striking of which is Fmp32. It is likely that all TA proteins are mis-localized to some low level but finding those that are particularly prone to this fate is a useful advance as it opens up analysis on natural substrates. Fmp32 might represent a protein whose abundance is regulated by this pathway.

The identification of Doa10 as an important factor in this pathway was judged to be clearly documented and convincing. However, the reviewers felt that the relationship between the Msp1/Doa10 pathway to the ERAD, GET pathway, and proteasome were not fully explored. In addition, the artificial dimerization experiment was felt to be somewhat incomplete. Finally, the reviewers felt that some of the negative data (i.e., the failure to find other factors) should be more fully and directly documented or interpreted with greater caution.

Essential revisions:

1) The authors make the sensible assumption that the requirement for Doa10 reflects its role in directly recognizing and ubiquitinating substrates for degradation by the proteasome. However, this is not documented and it seems to be an important aspect of their model worth investigating. Some additional information to verify that substrate ubiquitination is reduced in the absence of Doa10, and verification that degradation is primarily proteasome-mediated, would round out the evidence for their model.

2) The conclusion that Doa10 is the major ligase for degradation is based on the absence of finding other hits in the screen. However, negative data in high-throughput screens of this type need to be interpreted with caution. Thus, negative statements such as "We did not find […]" and "The only E3 ligase […]" are not warranted without more rigorous testing to verify that the desired genes for the process of interest (e.g., the full set of E3 ligases and adaptors) are really knocked out. If something like that has been done it should be described. Otherwise, the text should be written more carefully and avoid conclusions based on not finding something. It would be helpful to assess in the Discussion how many genes involved with ERAD were not assayed because they are essential, such as CDC48, or not included for some reason (such as NPL4). Given the direct relevance of ERAD to the process in question, the authors should directly test Hrd1, Cdc48 (hypomorph), and adaptors for comparison with their Doa10 result.

3) The rapamycin oligomerization experiment is insufficient to conclude that single (i.e., monomeric) TMDs are the key element for Msp1 recognition. This conclusion would be better supported if they made a version of one of their fusion proteins (either FKBP or FRB) lacking the Pex15 TMD. In this case, the rapamycin would still cause oligomerization, but there would only be one TMD in the oligomeric complex. Assuming this still gets degraded, the authors could confidently rule out an effect of rapamycin or cytosolic domain oligomerization. Related to such experiments (Figure 5C and Supplementary Figure 4A) it is not clear why the stabilization effect observed for GFP-FRB1-Pex15(TMD) upon the addition of rapamycin is not observed for the other partner in the dimer namely, HA-FKBP12- Pex15(TMD). The apparent effect in Figure 5C is also observed for the control protein Pgk1 and seems to be related to different loading. An explanation is needed to clarify this point.

---

## [Author Response]

Essential revisions:1) The authors make the sensible assumption that the requirement for Doa10 reflects its role in directly recognizing and ubiquitinating substrates for degradation by the proteasome. However, this is not documented and it seems to be an important aspect of their model worth investigating. Some additional information to verify that substrate ubiquitination is reduced in the absence of Doa10, and verification that degradation is primarily proteasome-mediated, would round out the evidence for their model.

We addressed this important point experimentally by pulldown experiments. On the one hand we immunopurified tFT-Pex15D30 from cells treated with proteasome inhibitor and could show that polyubiquitination is reduced in *doa10Δ* (Figure 4—figure supplement 1F). On the other hand, we performed immobilized-metal affinity chromatography of histidine tagged ubiquitin. These experiments confirm that poly-ubiquitination is reduced upon *doa10Δ* (Figure 4C). Furthermore, we show by cycloheximide chase experiments that proteasome inhibitors block degradation (Figure 3—figure supplement 1E). Our extended data, together with the recent publication by Li et al., 2019, investigating temperature sensitive yeast proteasome mutants in turnover of Pex15Δ30, corroborates our model.

2) The conclusion that Doa10 is the major ligase for degradation is based on the absence of finding other hits in the screen. However, negative data in high-throughput screens of this type need to be interpreted with caution. Thus, negative statements such as "We did not find…" and "The only E3 ligase…" are not warranted without more rigorous testing to verify that the desired genes for the process of interest (e.g., the full set of E3 ligases and adaptors) are really knocked out. If something like that has been done it should be described. Otherwise, the text should be written more carefully and avoid conclusions based on not finding something. It would be helpful to assess in the Discussion how many genes involved with ERAD were not assayed because they are essential, such as CDC48, or not included for some reason (such as NPL4). Given the direct relevance of ERAD to the process in question, the authors should directly test Hrd1, Cdc48 (hypomorph), and adaptors for comparison with their Doa10 result.

We agree that drawing conclusions from not finding additional factors in a given screen is difficult and therefore clearly point out this caveat by stating “that Ubc6, Ubc7 and Doa10 are the only significant hits”. Despite that, we have repeated our screen in a library including temperature sensitive alleles of otherwise essential E3 ligases and enriched for ubiquitin modifying enzymes corroborating our finding that Doa10 is the major E3 ligase. Overall, both our screen covered >85% of known E3 ubiquitin ligases (43 of 49, according to

Finley et al., 2012). Moreover, we now indicate (subsection “Doa10 controls TA abundance and protein targeting fidelity”) that key ERAD factors such as Cdc48 and Ubx2 were not tested because they are essential or were not included in the screened library. However, because annotations of the ERAD network may differ, we prefer not to indicate specific numbers in the text.

In order to support our interpretation that canonical ERAD is not involved in abundance control of Pex15Δ30, we have included cycloheximide chase experiments in *hrd1Δ* cells (Figure 3—figure supplement 1C), which do not show any effect. Moreover, we further strengthened our finding that the Doa10 complex acting on cytoplasmic pre-inserted TA proteins synergizes with mitochondrial Msp1 to ensure targeting fidelity.

3) The rapamycin oligomerization experiment is insufficient to conclude that single (i.e., monomeric) TMDs are the key element for Msp1 recognition. This conclusion would be better supported if they made a version of one of their fusion proteins (either FKBP or FRB) lacking the Pex15 TMD. In this case, the rapamycin would still cause oligomerization, but there would only be one TMD in the oligomeric complex. Assuming this still gets degraded, the authors could confidently rule out an effect of rapamycin or cytosolic domain oligomerization. Related to such experiments (Figure 5C and Supplementary Figure 4A) it is not clear why the stabilization effect observed for GFP-FRB1-Pex15(TMD) upon the addition of rapamycin is not observed for the other partner in the dimer namely, HA-FKBP12- Pex15(TMD). The apparent effect in Figure 5C is also observed for the control protein Pgk1 and seems to be related to different loading. An explanation is needed to clarify this point.

We followed this interesting idea and the concerns raised by generating new data on two soluble versions of the FKBP12 construct and clarification of some experimental shortcomings. As pointed out by the reviewers, we observe that fusion of FKBP12 to the Pex15 TM region stabilizes the protein construct. We are sorry that we have missed to indicate this limitation of our assay in the first submission. Since we now have confirmed this effect with a second fusion protein (mChe-FKBP12^TMD^), and similar effects have been reported by others (Edwards and Wandless, 2007; Morgan et al., 2014), we use this construct as mimic for a stable complex partner and restrict our interpretation to the FRB1 fusion protein that behaves like the initial reporter protein. Importantly, we now show data confirming that the FRB1^TMD^ reporter was degraded in an Msp1- and Doa10-dependent manner (Figure 6—figure supplement 1F). Furthermore, we added new data on two soluble versions of the FKBP12 construct. As shown in the Figure 6D and Figure 6—figure supplement 1H, we observed only a modest increase of the half-life compared to the complete stabilization when the membrane-anchored FKBP12^TMD^ was used (Figure 6B, C and Figure 6—figure supplement 1D). This result corroborates our initial interpretation that dimerization with a membrane integral protein counteracts Msp1 extraction and subsequent degradation of our Pex15-based reporter construct. Moreover, the increased cellular levels of FRB1^TMD^ by dimerization with a soluble FKBP12 constructs indicates that the Msp1-mediated extraction is influenced by energy required for unfolding and disassembly of the cytoplasmic client domain.